# Assembling of Metal-Polymer Nanocomposites in Irradiated Solutions of 1-Vinyl-1,2,4-triazole and Au(III) Ions: Features of Polymerization and Nanoparticles Formation

**DOI:** 10.3390/polym14214601

**Published:** 2022-10-29

**Authors:** Alexey A. Zharikov, Elena A. Zezina, Rodion A. Vinogradov, Alexander S. Pozdnyakov, Vladimir I. Feldman, Sergey N. Chvalun, Alexander L. Vasiliev, Alexey A. Zezin

**Affiliations:** 1Department of Chemistry, Lomonosov Moscow State University, 119991 Moscow, Russia; 2Favorsky Irkutsk Institute of Chemistry, Siberian Branch of the Russian Academy of Sciences, Favorsky St., 1, 664033 Irkutsk, Russia; 3Enikolopov Institute of Synthetic Polymeric Materials, Russian Academy of Sciences, Profsoyuznaya St., 70, 117393 Moscow, Russia; 4Moscow Institute of Physics and Technology, National Research University, 141701 Dolgoprudny, Russia

**Keywords:** kinetics of radiation-induced polymerization, gold nanoparticles, one-pot synthesis, metal–polymer nanocomposites, poly(1-vinyl-1,2,4-triazole)

## Abstract

Gold nanoparticles (AuNPs) stabilized with poly(1-vinyl-1,2,4-triazole) (PVT) have been synthesized via a one-pot manner in irradiated solutions of 1-vinyl-1,2,4-triazole (VT) and Au(III) ions. The transmission electron microscopy examinations have shown that the sizes of nanoparticles formed range from 1 to 11 nm and are affected by the ratio of VT to gold ions. To study the kinetics peculiarities of the VT polymerization and assembling of AuNPs, UV-Vis spectroscopy was used. The analysis of the data obtained reveals that an inhibition period, influenced by Au(III) concentration, is followed by the polymerization of a monomer. Importantly, the absorbed doses, corresponding to the onset of rapid polymerization, correlate with the doses at which the accelerated formation of AuNPs begins. The kinetics aspects, which could lead to such an effect, are discussed.

## 1. Introduction

Unique functional properties (optical, catalytic, electronic, etc.) of metal–polymer nanocomposites based on gold nanoparticles (AuNPs) have been stimulating interest in methods for their synthesis over the past few decades [1,2,3,4,5,6,7,8,9]. Actually, various approaches for the preparation of AuNPs stabilized by polymer matrices were developed; nevertheless, the general strategy is apparently based on the reduction in Au (III) ions with various chemical agents in the presence of a suitable polymer [9]. However, in recent years, great efforts have been paid to the development of methods for the production of nanocomposites, providing the synthesis of nanoparticles and the formation of a stabilizing matrix in “one pot” [10,11,12,13,14,15,16]. In this respect, specific attention has been focused on the radiation chemical approach, which can ensure the opportunities for obtaining both soluble nanocomposites and gels [14,15,16,17,18,19]. Furthermore, the radiation chemical method is a particularly attractive way to prepare high-purity nanocomposites without the use of a chemical reducing agent, since the irradiation of aqueous solutions leads to the generation of efficient and “clean” reducing species, such as hydrated electrons [20,21]. Likewise, the radiation-initiated polymerization allows one to obtain materials without residues of initiators or catalysts and makes it possible to carry out reactions at ambient temperature [15,16,17,19]. The adjustment of radiation parameters such as the radiation type (X- and γ-rays, or fast electrons), dose rate and total absorbed dose over a wide range provides great advantages for obtaining materials with versatile functional properties [20,21,22,23].

Poly(1-vinyl-1,2,4-triazole) (PVT), synthesized in this work by the radiation-induced polymerization of (1-vinyl-1,2,4-triazole) (VT), is a biocompatible, non-toxic and hydrophilic polymer, which also shows high thermal and chemical stability [24]. VT-polymers exhibit an excellent ability to stabilize gold and silver nanoparticles, which determines the benefits of application of the systems based on them for the preparation of metal–polymer nanocomposites [25,26,27]. Thereby, taking into consideration the above-mentioned benefits of PVT as a matrix for AuNPs, PVT-based nanocomposites might be promising materials for medical applications, including therapy and theranostics. Additionally, the opportunity to produce ultrafine colloids makes it possible to consider the prospects of catalytic applications of the PVT-protected AuNPs. Actually, small AuNPs ranging in size from one to several nanometers seem to be of particular interest owing to their extremely large total surface area. Recently, we reported that ultra-small AuNPs, with an average size of 1.5 nm, can be successfully obtained by the radiation chemical method in aqueous solutions in the presence of both PVT and poly(acrylic acid) [27].

Active species, generated in aqueous solutions of VT containing metal ions during irradiation, promote both the polymerization of a monomer and the formation of nanoparticles [16,17,19,20]. In this regard, the understanding of the mechanisms of the radiation-induced assembling of metal polymer nanocomposites is the key problem. The present work is mainly focused on the kinetic investigation of simultaneous VT polymerization and radiation chemical synthesis of AuNPs in one reactor, with specific attention being paid to the effect of the concentration of Au(III) ions on the efficiency of VT polymerization.

## 2. Materials and Methods

The starting monomer 1-vinyl-1,2,4-triazole was synthesized in the Favorsky Irkutsk Institute of Chemistry, Irkutsk, Russia according to the method described in [25]. The following reagents were employed for the preparation of samples: hydrogen tetrachloroaurate (III) trihydrate (Sigma-Aldrich, St. Louis, MO, USA), ethanol (analytical grade, Reachim, Moscow, Russia) and NaOH (0.5 M standard titer solution, Reachim, Moscow, Russia). Distilled water was used as a solvent. VT–Au(III) complexes (Table 1) were prepared as follows: an aqueous solution of HAuCl_4_ of the required concentration was added to an equal volume of 2 wt.% VT solution with continuous stirring. The pH was adjusted by adding sodium hydroxide. Before irradiation ethanol (10 vol.%) was added to the solutions as a scavenger of hydroxyl radicals. To remove air oxygen dissolved in a water–ethanol medium, the samples were bubbled with argon (extra pure grade) before and during irradiation. 

The samples were irradiated by X-rays using a 5-BKhV-6W tube with tungsten anode (applied voltage was 45 kVp, anode current ca. 80 mA) at 293 K in polypropylene Eppendorf test tubes of 8 mm thick. Such conditions ensured uniform generation of radiolysis products in the volume. The dose rate was 19 or 6.2 Gy/s, as determined using a ferrosulfate dosimeter irradiated in the same geometry as the samples studied. For metal–polymer complexes, the actual dose rate was calculated taking into account the mass absorption coefficients for X-rays [28] with an effective energy of 20 keV, which provides a reasonable representation of the actual X-ray spectrum.

The spectra of irradiated samples and turbidimetric data were obtained using a Perkin Elmer Lambda 9 spectrophotometer with an optical range of 200–900 nm (Perkin Elmer, Überlingen, Germany). in quartz optical cells (optical path 1 mm). The structure of the prepared nanocomposites was studied on a Tecnai Osiris FEI transmission microscope (FEI Company, Hillsboro, OR, USA) with a resolution of 1 Å. Samples for the TEM examinations were prepared by depositing a drop of the irradiated solution onto a Cu TEM-grid coated with a lacey carbon film (01844-F, Carbon Film only on 400 mesh, Cu, Tedpella, Redding, CA, USA) and then dried in an argon atmosphere. The specimens were stable under the electron beam and did not degrade during the observation.

## 3. Results

The radiation-induced polymerization of VT in the presence of Au(III) ions was studied at initial ratios of the monomer and Au(III) ions of 40/1, 80/1, 160/1 and 320/1. The samples prepared in these ratios are either transparent (80/1, 160/1, 320/1) or slightly opalescent solutions (40/1) and do not precipitate during irradiation. Meanwhile, an increase in the Au(III) concentration to the ratio of 20/1 leads to the gradual precipitation of the Au–VT complex, which greatly complicates kinetics investigations. Significantly, when mixing aqueous solutions of VT and HAuCl_4_ (the resulting pH is 2.5–3.0), a distinct band with the maximum at 310 nm (Figure 1A), corresponding to electronic transitions from occupied Cl^−^ ligand orbitals to unoccupied metal orbitals [29], practically disappears. The same effect was observed earlier for PVT-HAuCl_4_ aqueous solutions [27], and it clearly indicates the replacement of chlorine ions in the chloride complex by the triazole groups. The schematic representation of the proposed structures of the VT–Au(III) complexes is shown in Figure 1B. Depending on pH and the VT/Au(III) ratio, L could be either a low molecular weight ligand Cl^−^, OH^−^ or another VT molecule [27,29].

All experiments were carried out in a slightly acidic medium at an initial pH value of 5.6 (±0.2), which ensures the generation of reducing species with high yields [16,20] and prevents the hydrolysis of Au(III) ions [29]. Such conditions proved to be favorable both for the reduction in metal ions (Au(III), Ag^+^, Cu^2+^) in the presence of PVT and for the stabilization of nanoparticles by macromolecules [15,16,24,26,27]. 

The results obtained show that irradiation of the samples leads to a decrease in the absorption band with the maximum at 226 nm, which is assigned to the double bond (C=C) of the VT molecules [15,16] (Figure 2). The total absorbed doses corresponding to the complete conversion of VT into PVT are 5–7 kGy for samples prepared at the ratios of 40/1, 80/1 and 160/1. In the case of solutions with the lowest concentration of Au(III) (320/1), polymerization is virtually completed at a dose of 2–3 kGy. Simultaneously, initially colorless samples become either red (40/1 and 80/1) or brownish red (160/1), which qualitatively indicates the formation of AuNPs [3,27]. At the ratio of 320/1, the Au(III) concentration is too low; therefore, irradiation was not accompanied by a pronounced change in the color of the samples. The UV-Vis spectra of the irradiated samples are shown in Figure 3. The broad absorption bands at ca. 520 nm corresponding to the surface plasmon resonance (SPR) of AuNPs [3,27] rise when increasing the dose due to gradual conversion of Au(III) ions into AuNPs, up to doses of 5–10 and 15–20 kGy for samples with the ratios of 80/1 and 40/1, respectively. The “blue shift” of the position of the SPR bands maximum to the UV region seems to be associated with an increase in the electron density in the metal due to the reduction in Au ions adsorbed on the AuNPs surface [30].

For TEM analysis, the samples with the ratio of 40/1 and 80/1, irradiated to a dose of 14 kGy, were used. We also have examined the sample with the ratio of 40/1 irradiated to a dose of 23 kGy, since its spectrum (Figure 3, 40/1, Spectrum 6) differs in the width of the SPR band and the position of the maximum from Spectrum 5 corresponding to the VT–Au(III) solution irradiated to a dose of 14 kGy. The bright field (BF) TEM image, together with the selected area electron diffraction pattern (SAEDP), is shown in Figure 4. The analysis of microdiffraction patterns indicates the presence of rings corresponding to the interplanar distances 2.35, 2.03, 1.44 and 1.23 Å (Figure 4D), which, according to the data [31], fit perfectly to the interplanar distances of gold crystals. The size distribution of the AuNPs obtained is given in Figure 5. First, it should be noted that the application of 1 wt.% VT solutions over a wide range of Au(III) concentrations (Table 1) makes it possible to obtain quasi-spherical (isotropic) AuNPs, the sizes of which do not exceed 12 nm. This fact indicates the strong stabilizing ability of PVT with respect to gold colloids due to the effective interaction of triazole groups with the metal surface. However, the size distributions of the AuNPs formed (Figure 5B,C) does not exhibit pronounced maxima. Thus, an increase in the Au(III) concentration leads to a broadening of the size distribution of the resulting nanoparticles; nevertheless, the average sizes remain in the range of 3–6 nm (Figure 5B,C). It is noteworthy that irradiation of the samples with the ratio of 40/1 to a dose of 23 kGy results in some decrease in the fraction of AuNPs with sizes of 2–3 nm, occurring simultaneously with an increase in the fraction of relatively large nanoparticles (8–11 nm). The latter fact can also be confirmed by analyzing the UV-Vis spectra of irradiated solutions (Figure 3, 40/1, Spectra 5 and 6); the “sharpness” of the AuNPs’ SPR band (Figure 3, 40/1, Spectrum 6) indicates the generation of lager nanoparticles [3]. In turn, the broadening of the band (Figure 3, 40/1, Spectrum 5) could mean the presence of the aggregates of closely spaced small nanoparticles around a large one, which coalesce upon further irradiation. Indeed, the formation of such aggregates at the final stages of the reduction in gold ions can be facilitated by the significant enhancement of local dose rate and the rate of the radiation chemical reduction in Au(III) ions in the vicinity of pre-existing nanoparticles. Such an effect, referred to as “radiation chemical contrast”, was previously described for swollen films of interpolyelectrolyte complexes with copper and silver, irradiated with X-rays [32]. It was attributed to an increasing local absorbed dose rate due to enhanced secondary electron emission from metal nanoparticles having much higher X-ray absorption cross-section in comparison with the surrounding medium [32]. 

## 4. Discussion

The active species formed in aqueous-alcohol solutions upon irradiation play the major role in the processes of radiation-induced polymerization and assembling of the metal nanoparticles [20]. Given the fact that water constitutes at least 90 wt.% of the sample composition, it would be reasonable to consider that X-rays are mainly absorbed by water. Nevertheless, the concentrations of ethanol and VT are rather high; thus, one should bear in mind the possibility of direct action of X-rays on ethanol and VT molecules, leading to the formation of additional primary radicals. The generation of the key products of water radiolysis at the homogeneous stage (pH 4–9) can be represented according to a well-known scheme (reaction 1) [20]:H_2_O → *e*_aq_, ·OH, H·, H_2_, H_2_O_2_, H_3_O^+^(1)

The hydrated electrons and ·OH radicals formed in the highest yields possess respectively strong reducing (*E*^0^ = −2.9 V_NHE_ [33]) and oxidizing properties (*E*^0^ = 1.9 V_NHE_ [33]). Thus, if the former species acts as the main reducing agent for metal ions [20,21], then the latter can oxidize isolated metal atoms and ions in intermediate valences [20,21]. In order to provide favorable conditions for the formation of metal nanoparticles, the ·OH radicals should be scavenged, for instance, with aliphatic alcohols [16,20]. However, at the same time, the ·OH radical acts as an initiator of radical polymerization [16,17]. In our recent work, we established that the presence of ethanol (10 vol.%) has no noticeable effect on the radiation chemical yield of the loss of the monomer upon irradiation of 1 and 10 wt. % VT aqueous solutions [16]. This fact indicates a comparable efficiency of chain initiation by both ·OH and alcohol radicals formed by the reactions (2, 3):CH_3_CH_2_OH + ·OH → CH_3_·CHOH + H_2_O(2)
CH_3_CH_2_OH + ·H → CH_3_·CHOH + H_2_(3)

Nevertheless, ethanol significantly affects the molecular weight distribution of the target polymer product, leading to a decrease in a high weight fraction, probably due to the chain transfer reaction (4) [16]:R(M)_n_M · + CH_3_CH_2_OH → R(M)_n_MH + CH_3_·CHOH(4)

It is worth noting that, unlike alcohol radicals, hydroxyl radicals can abstract hydrogen atoms from macromolecules, which again results in the formation of macroradicals (reaction 5) capable of subsequent chain propagation and crosslinking of macroradicals due to recombination (reactions 6, 7) [16]:OH + R(M)_n_ → R(M)_n-1_M + H_2_O(5)
R(M)_n-1_M_˙_ + M → R(M)_n_M(6)
R(M)_m_M· + R(M)_n_M → R(M)_n+m+2_R(7)

Obviously, these reactions are responsible for the subsequent increase in molecular weight in the absence of alcohol. Moreover, another important role of ethanol is concerned with ensuring a reducing environment during irradiation. CH_3_·CHOH radicals possess potential *E*^0^ (CH_3_CHO, H^+^/CH_3_·CHOH) of −1.1 V_NHE_ [33] and, being oxidized, yield acetaldehyde, which is known to be a mild reducing agent. Thus, it should be emphasized that CH_3_·CHOH radicals may serve both as initiators of polymerization and as a reducing agent for Au(III) ions upon irradiation of VT aqueous-ethanol solutions containing Au(III).

The results obtained clearly reveal the effect of Au(III) ions on the kinetics of VT polymerization. Indeed, according to the data shown in Figure 6, the presence of Au(III) ions (the ratio of VT to Au(III) of 160/1) leads to a certain decrease in the rate of monomer conversion, while the samples with the ratio of 80/1 and 40/1 show a pronounced inhibition period. The effective radiation chemical yields (*G*(-VT)) of the loss of monomer (see Table 2) were estimated by linear approximation of the initial sections (the number of experimental points corresponding to the selected sections indicated in parentheses in Table 2) of the dependencies shown in Figure 6, according to the formula (8):
(8)$$G\left(-\mathrm{VT}\right)=\frac{9.65\times 10^{6}f}{\rho \varepsilon l}\times \frac{\Delta A}{D}$$
where *A* is the optical density at wavelength of 226 nm, which is assigned to the maximum of the absorption band of the VT double bond, *D* is the absorbed dose in Gy, *ρ* is the density of irradiated solutions (in each case, approx. to 1 g/cm^3^), *ε* is the molar extinction coefficient of VT at 226 nm (1.1 × 10^4^ L⋅mol^−1^⋅cm^−1^), *l* is the path length (0.1 cm) and *f* is the dilution factor (50). The effective radiation chemical yields calculated in this way should be considered as effective values that make it possible to compare the rates of VT conversion at different concentrations of Au(III) ions. Thus, as one can infer from Figure 6 and the data displayed in Table 2, the Au(III) ion acts as an inhibitor of the radical polymerization of VT. Actually, despite the sufficiently low concentration of Au(III), and especially in the case of solutions with the ratio of 320/1, a certain decrease in the radiation chemical yield compared to that without Au(III) is still observed. Meanwhile, in the case of radiation-induced polymerization of VT in the presence of Ag^+^, the situation appears to be different. As shown previously [16], for samples with the ratio of VT to Ag^+^ equal to 25/1, no inhibition period is found, although there is some decrease in the radiation chemical yield *G*(-VT), which may be explained by the competition between the reactions of alcohol radicals with monomer molecules and Ag^+^ ions adsorbed on the metal surface.

Another important point in kinetic studies is concerned with the development of the absorption band at ca. 520 nm during irradiation. The data presented in Figure 7 clearly demonstrate that a long induction period is followed by an accelerated formation of AuNPs. Furthermore, the induction dose rises when increasing the Au(III) concentration. Significantly, the absorbed doses, corresponding to the onset of rapid polymerization, correlate with the doses at which the accelerated formation of AuNPs begins (Table 2). Thereby, along with the sigmoidal (S-shaped) kinetics of nanoparticle growth [34,35], an “S-reverse” kinetics of the monomer consumption takes place (Figure 8).

In order to explain the observations described above, we should now consider the mechanism of the radiation-induced reduction in Au(III) and the formation of AuNPs. Briefly speaking, the generation of isolated Au^0^ atoms, being the nuclei of growing clusters, occurs via a multistep reduction [35,36]. An important feature of this mechanism is related to the essentially negative reducing potential of an isolated zerovalent gold atom [35,36,37]. As has been proposed in the work [36], Au^0^ are able to reduce Au(III) ions into Au(II), thus preventing the appearance of gold clusters. In other words, until the reduction in Au(III) and the accumulation of Au(I) are completed, the formation of gold clusters (Au_n_) does not occur. In turn, this results in a relatively long induction period, during which no absorption band corresponding to the LSPR of (Au_n_) clusters is observed. The simplified scheme of the radiation chemical transformation of Au(III) ions, proposed in our case, can be represented as follows (in our nomenclature, L means a low molecular weight ligand, either OH^−^ or Cl^−^ [29]):

(I) Induction period:[Au^III^(VT)_k_(L)_4−k_]^(1−k)^− + CH_3_·CHOH → [Au^II^(VT)_k_(L)_4−k_]^(2−k)−^ + CH_3_CHO + H^+^(9)
[Au^III^(VT)_k_(L)_4−k_]^(1−k)−^ + *e*_aq_ → [Au^II^(VT)_k_(L)_4−k_]^(2−k)^(10)
2[Au^II^(VT)_k_(L)_4−k_]^(2−k)−^ → [Au^III^(VT)_k_(L)_4−k_]^(1−k)−^ + [Au^I^(VT)_m_(L)_2-m_]^(1−m)−^ + (2 − *k* + *m*)L^−^ + (*k* − *m*)VT(11)
[Au^I^(VT)_m_(L)_2-m_]^(1−m)−^ + *e*_aq_ (CH_3_·CHOH ?) → (Au^0^) _VT, L_(12)

(II) Accelerated formation and growth of nanoparticles:Au^0^ + Au^I^ → Au_2_^+^ → … → (Au_n_, Au^I^_m_)^m+^ _clusters_(13)
(Au_k_, Au^I^_l_)^l+^ + (Au_m_, Au^I^_n_)^n+^ → (Au_k+m_, Au^I^_l+n_)^m+^(14)
(Au_n_, Au^I^_m_)^m+^ _clusters_ + CH_3_·CHOH (*e*_aq_) → (Au_n+1_, Au^I^_m−1_)^(m−1)+^ _clusters_(15)

Reactions 9–11 describe the major processes occurring during the induction period. As mentioned above, ethanol radicals can effectively reduce Au(III) ions, resulting in Au(II) ions, which are disproportionate, apparently via the formation of relatively long-lived dimers (Au^II^)_2_ [36]. Thereby, the Au(I) ions, being the most stable intermediate low-valence product of Au(III) reduction, gradually accumulate during the initial irradiation time. The next stage (reaction 12) involves the appearance of isolated solvated gold atoms (Au^0^)_L_ followed by a burst association into small clusters and their coalescence (reactions 13, 14). These reactions represent the nucleation step. The formation of isolated gold atoms requires a strong reducing agent, since the standard reduction potential *E*^0^ of the Au^+^/Au^0^_isolated atom_ pair is much lower than that of the Au^+^/Au_bulk_ pair (+1.69 V_NHE_) [38]. In fact, to estimate the former potential, one should take into account the sublimation free energy of gold, and thus, we can get a value of −1.5 V_NHE_ [37]. However, this value does not include corrections associated with the formation of Au(I) complexes with VT and the solvation of the gold atom, which can strongly affect the redox potential. Thus, for example, the standard potentials *E*^0^ corresponding to the reduction of some complex silver ions to solvated atoms are significantly less [39,40] than the standard potential *E*^0^ (−1.8 V_NHE_) of the Ag^+^(H_2_O)/Ag^0^(H_2_O) pair [41]. Therefore, the potential of the Au^+^/Au^0^_isolated atom_ pair should be considered as a rough estimate, while the real redox potential *E*^0^ (Au^+^)_L,VT_/(Au^0^)_L,VT_ has to be more negative. Consequently, from the point of view of thermodynamics, ethanol radicals are not capable of reducing Au(I) ions to isolated atoms (reactions 12) and, as a result, this reaction can be provided only by a hydrated electron. On the contrary, the reduction in adsorbed Au(I) ions taking place on the metal surface (reaction 15) may occur due to reducing agents having a lower reduction potential, including acetaldehyde.

Within the framework of the mechanism describe above, the observed features of the polymerization kinetics can be understood in terms of the competition between Au(III) ions and VT molecules for ethanol radicals. When adding 10 vol.% of ethanol, the overwhelming majority of initiation processes should be triggered by the reactions involving alcohol radicals [16]. However, despite the rather high concentration of VT compared to Au(III), the ratio (16) could be significantly less than 1 due to the large difference in the rate constant *k*_Au_ and *k_VT_* corresponding to reactions 9 and the reaction of the addition of ethanol radical to VT (Figure 1), respectively.
(16)$$r=\frac{k_{VT}\left[VT\right]}{k_{Au\left(III\right)}\left[Au\left(III\right)\right]}\ll 1$$

Indeed, it is known that the rate constants of the reactions of carbon-centered radicals’ addition to double bonds, in many cases, do not exceed 10^4^ M^−1^s^−1^ [42]. Thus, for example, the rate constant of the reaction of a methyl radical with ethylene (a carbon-centered radical and an alkene with the simplest structure) in liquid phase is of 7 × 10^3^ M^−1^s^−1^ [42]. Although OH and triazole groups significantly affect the electron density in the radical and alkene, respectively, which, in turn, affects the activation energy and rate constant, that might increase by two or three orders of magnitude. On the other hand, steric hindrances associated with the ethanol radical and VT molecule should play against the increase in the reaction rate constant *k*_VT_. At the same time, the rate constant *k*_Au(III)_, apparently, could be close to the diffusion-limited value, that is, at least three orders of magnitude higher than *k*_VT_. Indeed, for example, the scavenging rate constant of isopropanol radicals with Au(III) ions in 2-propanol was reported to be 7.6 × 10^9^ M^−1^s^−1^ [36]. Therefore, the difference in the rate constants *k*_VT_ and *k*_Au(III)_ probably can account for the observed induction period of polymerization directly associated with the reduction in Au(III) ions, which was revealed in the experiment. In turn, at the stage of formation of isolated atoms and small clusters, alcohol radicals do not play the role of a reducing agent; instead, they initiate rapid polymerization processes. Furthermore, according to this explanation, the initial rate of VT conversion should increase if ·OH radicals would initiate the polymerization. To confirm this assumption, we also performed a series of experiments without ethanol (the samples with the VT/Au(III) ratio of 80/1), the results of which are shown in Figure 9. Indeed, the radiation chemical yield of the chain polymerization process monitored by observation of the loss of VT molecules in the absence of ethanol increases by seven times compared to that in the presence of ethanol (570 and 80 molecules per 100 eV, respectively). However, even for the ethanol-free solutions, the efficiency of VT conversion does not reach the value observed in the absence of Au(III) ions (2.3 × 10^3^ molecules per 100 eV [16]). Presumably, this effect is associated with the reduced initiation rate due to partial consumption of ·OH radicals by gold ions in lower valencies, which were formed due to the reduction in Au(III) ions by hydrated electrons [35]. 

Finally, we should make some notes about PVT, as a matrix for AuNPs. It is known that the stabilizing efficiency of diverse polymers is determined by the character of the interaction between macromolecules and nanoparticles. Moreover, the difference in the adsorption energy of polymers’ functional groups on specific crystallographic facets may ensure preferential growth along certain crystal directions, and, thus, promote the sharp-controlled formation of metal nanoparticles [43,44]. For instance, using *β*-cyclodextrin affects the final morphology of silver nanoparticles, leading to the selective growth of anisotropic AgNPs [44]. Another example is concerned with the shape-controlled synthesis of AgNPs in temperature-responsive grafted polymer brushes [43]. Meanwhile, polymer stabilizers such as PVT, possessing a strong affinity for the metal precursor and metal surface [16,26,27], would facilitate the formation of relatively small and quasi-spherical nanoparticles. PVT, which is a nonionogenic amphiphilic polymer with hydrophilic functional groups and a hydrophobic backbone, is an effective stabilizing matrix for AuNPs, due to the high ability of functional groups to interact with the gold surface. Triazole groups, being strong ligands for the Au(III) ions, can be adsorbed on the metal surface and, thus, provide steric stabilization [27] of nanoparticles. In addition, hydrophobic interactions of backbones also can contribute to the stabilizing effect of colloids [45]. Therefore, the PVT–AuNPs’ dispersions show high colloidal stability and do not aggregate for at least a week, which can be monitored by the SPR spectra of AuNPs (see Appendix A).

## 5. Conclusions

It has been found that Au(III) ions affect the polymerization of VT; namely, an increase in the initial concentration of Au(III) leads to the inhibition of the VT’s conversion into a polymer. Moreover, it has been revealed that intense polymerization and the stage of nanoparticles nucleation occur in the same range of absorbed doses. This fact suggests interplay between the processes of assembling nanoparticles and the formation of a polymer matrix. A logical explanation of this effect is concerned with the competition of the reactions of Au(III) reduction and the initiation of polymerization. Indeed, in the presence of 10 vol.% of ethanol, these processes are triggered by alcohol radicals, which act both as a polymerization initiator and as a reducing agent for Au(III) ions. Since ethanol radicals are apparently unable to reduce Au(I) ions into isolated atoms, at the nucleation stage, they act primarily as initiators, which leads to an acceleration of the VT polymerization. Notably, in the case of the radiation-induced formation of AgNPs in irradiated solutions containing silver ions and VT, no inhibition of polymerization was observed [16], which is explained by the fact that alcohol radicals cannot participate in the reduction in Ag^+^ ions until the metal surface is formed. 

To summarize, we have to stress that the results obtained show the prospects of the one-pot method for the synthesis of the PVT-based metal–polymer nanocomposites containing AuNPs. Similarly to the case of AuNPs assembling in aqueous PVT solutions, the formation of colloids in VT solutions occurs under conditions of strong interactions of triazole groups with the metal surface, resulting in obtaining relatively small nanoparticles. Significantly, the radiation chemical approach makes it possible to carry out the reduction in metal ions and polymerization of a monomer in one reactor and provides for obtaining pure nanocomposites without the use of any chemical reagents, which is important for the development of “green” methods to produce the AuNPs. Thereby, the PVT-stabilized AuNPs can be considered promising composite materials for medical, biotechnological and catalytic applications.

## Data Availability

Not applicable.

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
