# Peer review of "Assembling of Metal-Polymer Nanocomposites in Irradiated Solutions of 1-Vinyl-1,2,4-triazole and Au(III) Ions: Features of Polymerization and Nanoparticles Formation"

_polymers, 2022, doi:10.3390/polym14214601_

Round 1
Reviewer 1 Report
The paper presents interesting results on the fabrication of gold nanoparticles stabilized with poly(1-vinyl-1,2,4-triazole) using a radiation-induced method. The paper can be accepted for publication after major revision. The following issues should be clarified:
It is a well-known fact that during the fabrication of metallic nanoparticles in a polymer matrix, the complexation between monomer units and ions plays a key role. The hypothetical representation of the complex of 1-Vinyl-1,2,4-triazole with HAuCl4 should be added.
What about the shape of the obtained nanoparticles? In general, the chemical nature of the polymer has a strong impact on the shape of synthesized metallic nanoparticles (https://doi.org/10.1016/j.apsusc.2018.09.033; https://doi.org/10.1021/jp200662j). Appropriate discussion should be presented.
The obtained assembly of metal-polymer nanocomposites has strong potential for different applications. Appropriate discussion on this topic should be provided.
Please cite the above-mentioned papers:
https://doi.org/10.1016/j.apsusc.2018.09.033
https://doi.org/10.1021/jp200662j
Reviewer 2 Report
The manuscript is well argued and written. It makes important contributions for the development of “green” methods to produce the AuNPs. I recommend its publication and a series of minor corrections.
In section 2. Materials and Methods, more information is needed related to the techniques used to characterize the obtained nanocomposites.
On page 9, line 13, I suggest that the mathematical expression describing the reaction rate be numbered, just like the other equations in the manuscript.
A more extensive characterization of the stability of the nanocomposites presented in this manuscript would add value to the authors' work.
Author Response
Please see the attachement

Round 2
Reviewer 1 Report
The authors have answered all of my issues and the paper can be accepted in its present form.